# Nanosystems, Drug Molecule Functionalization and Intranasal Delivery: An Update on the Most Promising Strategies for Increasing the Therapeutic Efficacy of Antidepressant and Anxiolytic Drugs

**DOI:** 10.3390/pharmaceutics15030998

**Published:** 2023-03-20

**Authors:** Jéssica L. Antunes, Joana Amado, Francisco Veiga, Ana Cláudia Paiva-Santos, Patrícia C. Pires

**Affiliations:** 1Department of Pharmaceutical Technology, Faculty of Pharmacy, University of Coimbra, 3000-548 Coimbra, Portugal; 2REQUIMTE/LAQV, Group of Pharmaceutical Technology, Faculty of Pharmacy, University of Coimbra, 3000-548 Coimbra, Portugal; 3Health Sciences Research Centre (CICS-UBI), University of Beira Interior, Av. Infante D. Henrique, 6200-506 Covilhã, Portugal

**Keywords:** anxiety, blood–brain barrier, brain bioavailability, depression, drug molecule functionalization, intranasal, nanoemulsions, nanoparticles, nanosystems, nose-to-brain

## Abstract

Depression and anxiety are high incidence and debilitating psychiatric disorders, usually treated by antidepressant or anxiolytic drug administration, respectively. Nevertheless, treatment is usually given through the oral route, but the low permeability of the blood–brain barrier reduces the amount of drug that will be able to reach it, thus consequently reducing the therapeutic efficacy. Which is why it is imperative to find new solutions to make these treatments more effective, safer, and faster. To overcome this obstacle, three main strategies have been used to improve brain drug targeting: the intranasal route of administration, which allows the drug to be directly transported to the brain by neuronal pathways, bypassing the blood–brain barrier and avoiding the hepatic and gastrointestinal metabolism; the use of nanosystems for drug encapsulation, including polymeric and lipidic nanoparticles, nanometric emulsions, and nanogels; and drug molecule functionalization by ligand attachment, such as peptides and polymers. Pharmacokinetic and pharmacodynamic in vivo studies’ results have shown that intranasal administration can be more efficient in brain targeting than other administration routes, and that the use of nanoformulations and drug functionalization can be quite advantageous in increasing brain–drug bioavailability. These strategies could be the key to future improved therapies for depressive and anxiety disorders.

## 1. Introduction

### 1.1. Pathophysiology and Treatment of Depression and Anxiety Disorders: Current Aspects and Limitations

Anxiety is a central nervous system (CNS) disorder characterized by tension, restlessness, and increased effort to concentrate, with a persistent depressed mood and lack of interest in activities which pleasure was once taken from. We can consider the existence of several types of anxiety disorders, such as disorders related to separation, social anxiety, panic, specific phobias, and generalized anxiety disorder. These disorders generally begin in childhood, adolescence, or early adulthood. The general physiological mechanism by which these types of disorders arise is related to the γ-aminobutyric acid (GABA), which is inhibited and, hence, does not fulfill its role in the downregulation of neuronal excitability. It is estimated to affect millions of people worldwide, leading to a great loss in a person’s quality of life [1,2,3,4,5,6,7].

In turn, depression is the biggest public health problem, making it a common mental disorder and one of the world’s leading causes of disability. In 2020, this disease affected about 16% of the world’s population [8]. The etiology of depression has been related to stress. It is usually manifested by a loss of interest, feelings of guilt, depressed mood, sleep disturbance, low energy, and suicidal thoughts and attempts. There are several theories around the origin of depression, but the most accepted is the monoamine theory, which is related to a decrease in serotonin, noradrenaline, and dopamine levels in the CNS. It is caused by decreased excitability on the dopaminergic and/or serotonergic pathway. The markers of oxidative stress may also indicate depression, such as low levels of glutathione (GSH) and other antioxidants, and high levels of thiobarbituric acid (TBARS), F2 isoprostanes, inflammatory cytokines, and reactive oxygen species. Depression has also been associated with low levels of catalase, an enzyme responsible for the degradation of hydrogen peroxide into water and oxygen. In depressive situations, this enzyme is in deficit and, therefore, there is an accumulation of reactive oxygen species and consequently oxidative stress [6,9,10,11,12,13,14,15,16,17].

Nowadays, oral and intravenous (IV) administrations are the most used in patients with depression or anxiety. However, for drugs whose target is the CNS, these routes have many disadvantages. The oral administration of central-acting drugs results in a low drug uptake by the brain and high drug distribution in the peripheral tissues. One of the main reasons for the failure of antidepressants and anxiolytics is the presence of the blood–brain barrier (BBB) and the existence of efflux pumps in the brain capillaries, endothelial cells, luminal membranes, and caveolae. As most antidepressant and anxiolytic drugs are substrates of these transporters, their brain bioavailability is limited, which causes a decrease in their effectiveness. Additionally, drugs that are administered orally can undergo chemical and metabolic degradation in the gastrointestinal tract, have drug–drug or food–drug interactions, have a slower therapeutic action which makes them non-adequate for emergency situations, and are only suitable for patients with the ability to swallow. On the other hand, the IV route is best in cases of emergency or when the patient is unable to swallow. However, it also has many disadvantages, such as invasiveness and the need for patient hospitalization, and difficulties in delivering the drug to the CNS due to the lack of penetration of the BBB [12,14,18,19,20,21].

The BBB is a physical and metabolic barrier that limits the transport of substances between the blood and the neuronal tissue and is responsible for maintaining the physiological stability of the brain and protecting the CNS from toxic agents and microorganisms. It is made of three layers, but it is the innermost layer that poses a greater problem for drug delivery to the CNS. The BBB is essentially made up of endothelial cells in the capillary walls and tight junctions that prevent the transport of drugs through the paracellular pathway between adjacent endothelial cells in the inner layer. The BBB also has a biochemical layer with high levels of efflux transport proteins, such as P-glycoprotein (P-gp) and multidrug-resistant protein-1, as well as the expression of many metabolic enzymes, which limit brain-drug uptake [22,23]. Despite all these limitations, small (molecular weight below 500 Da) and hydrophobic molecules, and some cells (such as monocytes, macrophages, and neutrophils) can be selectively transported to the brain [21,22,23,24,25].

Additionally, although currently a lot of different treatment strategies exist, including pharmacological treatments, psychotherapies, and brain stimulation techniques, less than half of patients achieve a complete remission with the first treatment [17,26,27]. For these reasons, several solutions have been studied to fight depression and anxiety in a more effective and safe way.

### 1.2. Potential Strategies for Enhancing Brain Drug Targeting and Bioavailability

#### 1.2.1. Intranasal Administration

The nasal cavity has three main regions: the vestibular, respiratory, and olfactory regions. The vestibular region measures approximately 0.6 cm^2^ and is located directly at the entrance of the nostrils. It contains nasal hairs (responsible for filtering inhaled particles), squamous epithelial cells, and some ciliated cells [28,29,30,31]. Next to it is the respiratory region (Figure 1), which corresponds to the largest nasal area, at 150 cm^2^ [21,28]. It is the most vascularized region with the greatest variety of cells, containing goblet, ciliated, non-ciliated, and basal cells [28,31,32]. Goblet cells are responsible for secreting mucin, water, salts, a small group of proteins and lipids, and, together with some nasal glands, form the mucus layer. The mucus forms a layer in the respiratory epithelium and can trap inhaled molecules and send them to the pharynx, where they pass through to the gastrointestinal tract [21,28,29]. Basal cells are the key cells of the nasal cavity and have the ability to differentiate into another type of epithelial cell, if necessary. These cells also help to attach the ciliated and goblet cells to the lamina propria [21,29]. Ciliated cells, as the name suggests, have cilia that increase their surface area. They help to move the mucus toward the nasopharynx, resulting in mucociliary clearance [21,32]. Together with a high degree of vascularization, this makes the nasal respiratory region a site of high drug transport into the systemic circulation. However, the respiratory region is also innervated by the maxillary and ophthalmic branches of the trigeminal nerve (V1, V2), which originate in the brainstem fossa and have been suggested as a possible target for drug transport to the CNS [28,30].

In turn, the olfactory region is made up of olfactory receptors, the olfactory epithelium, and lamina propria (Figure 2). Olfactory receptors are unmyelinated neurons located in the nasal epithelium. In the lamina propria, each olfactory receptor forms thick bundles of axons that become olfactory nerves and innervate the cribriform plate, forming synaptic connections with the glomeruli of mitral cells in the olfactory bulb. There are two types of basal cells in this region: horizontal basal cells and spherical basal cells. In addition to horizontal basal cells, multipotent progenitor cells, and globular basal cells, this region also contains structural support cells that encase the olfactory receptors in the olfactory region, maintaining the structural integrity and ionization of the olfactory receptors. Drugs with small sizes can be transported through the axons, by the olfactory bulb, to the olfactory cortex, reaching the cerebellum [21,32,33].

Nasal secretions are composed of approximately 95% water, 2% mucin, 1% salts, 1% other proteins (albumin, immunoglobulins, lysozymes, and lactoferrin), and <1% lipids. They move through the nose at a rate of approximately 5 to 6 mm/min, resulting in particles being cleared from the nose every 15 to 20 min. In addition, enzymes such as isoforms of the cytochrome P450 (CYP1A, CYP2A, and CYP2E), carboxylesterases, and glutathione S-transferases can also be found in the nasal cavity Therefore, the residence time of the formulation (and, consequently, the drug) in the nasal cavity will also be affected by these enzymes’ metabolism. To prolong the residence time of the formulation in the nasal cavity, and consequently increase the amount of drug that will be absorbed, biologically adhesive (mucoadhesive) excipients such as gelatin, chitosan, carbopol, and cellulose derivatives can be used [28,34].

IV drug delivery to the brain is largely influenced by the drug’s plasma half-life, the extent of the metabolism, the degree of non-specific binding to plasma proteins, and the permeability of the compound across the BBB and into the peripheral tissues. Due to these issues, intranasal (IN) administration has been presented as a promising alternative to the IV route, having gained increasing interest over the past few years. After IN administration, the drug can take three different routes to the brain: one intracellular route and two extracellular routes (Figure 2 and Figure 3). The intracellular route is activated after drug endocytosis by the olfactory sensory cells, and the drug is consequently transported through the neuronal axon to the synaptic clefts in the olfactory bulb, where it is released through exocytosis. Drugs are delivered by endocytosis to the olfactory sensory neurons and peripheral trigeminal neurons, and then transported intracellularly from the olfactory sensory nerve to the olfactory bulb, and from the trigeminal nerve to the brainstem. This is an extremely slow pathway and it can take hours for the drug to reach the olfactory bulb [28,33,35,36]. In the extracellular mechanism (Figure 3), the drug can follow two different routes: it can cross the gaps between the olfactory neurons and then be transported to the olfactory bulb, or it can be transported along the trigeminal nerve to avoid the BBB. Once the drug has reached the olfactory bulb or the trigeminal region, it can be transported to other areas of the brain by diffusion, which is facilitated by a perivascular pump that is activated by arterial pulsation [28,32,35].

#### 1.2.2. Nanosystems

The currently marketed nasal preparations exist mainly in the form of solutions or suspensions, but labile, poorly soluble, poorly permeable, and/or less potent drugs may require a formulation that promotes drug bioavailability and, preferably, direct delivery to the brain. To this end, nanotechnology has emerged to fill the existing gaps. All nanosystems are characterized by their small size, which makes them suitable for transporting drugs to the target tissues and cells, where they are ideally released [20,37,38].

In terms of nanosystems that are meant to be administered by the IN route, these are designed to provide a longer residence time in the nasal cavity, overcome nasal mucociliary clearance, and facilitate rapid drug transport across the nasal mucosa. Independently of the intended administration route, there are several types of nanosystems (Figure 4), with the main being: polymeric nanoparticles (NP), which are divided into nanocapsules and nanospheres; lipid nanoparticles, namely solid lipid nanoparticles and nanostructured lipid carriers; liposomes; nanometric emulsions, such as nanoemulsions and microemulsions; and nanoemulgels [20,38,39,40,41].

##### Polymeric Nanoparticles

Polymeric nanoparticles are compact colloidal systems with a variable size range within the nanometric scale and can be composed of natural or synthetic polymers. They are composed of a dense core of polymeric matrix, suitable for encapsulating lipophilic drugs, and a hydrophilic crown that provides stability to the nanoparticles. The drug can be incorporated into the nanosystem in several ways: dissolved in the matrix, encapsulated within it, or adsorbed to it [20,39,42].

There are two main types of polymeric NP: nanocapsules (a reservoir system) and nanospheres (a matrix system). Nanocapsules consist of an oily core in which the drug is dissolved, surrounded by a polymeric shell that controls the drug release from the core. Nanospheres are based on a continuous polymeric network, and in these systems the drug is either retained inside the nanosphere or adsorbed onto its surface. Although NP can be presented in a variety of ways, they have common characteristics that make them advantageous formulations, such as biocompatibility, biodegradability, high drug loading, the possibility of controlled drug release, and stability during storage [43,44,45].

Several polymers can be used to produce NP for drug delivery. One of these is chitosan, a linear, natural, and biocompatible polysaccharide derived from the deacetylation of chitin from crustacean shells. It is a polymer that has some of the properties we need for effective and direct drug delivery to the brain, since it has the capacity to reduce mucociliary clearance and transiently opens tight junctions (protein kinase C pathway interaction), facilitating paracellular drug transport across the nasal mucosa to the brain [21,35,36,46]. Chitosan is insoluble at neutral and basic pH values, but forms salts with inorganic or organic acids, such as hydrochloric acid and glutamic acid, which are soluble in water up to about pH 6.3 (depending on the molecular weight and deacetylation degree and, therefore, pKa). To further improve the mucoadhesive properties of chitosan, derivatives with thiol groups have been developed. These groups provide enhanced ciliary mucoadhesion, by forming covalent bonds between the polymer and the mucus layer that are stronger than non-covalent bonds [21,35,47].

Another polymer of interest is alginic acid, a natural polysaccharide found in the cell walls of brown algae. Alginate (AG) is a derivative of alginic acid, and it is usually used in formulations in salt form (sodium or calcium). It is a hydrophilic polymer that, when in the presence of divalent cations, forms a gel that allows a controlled drug release [21,46,48].

Poly lactic-co-glycolic acid (PLGA) is a synthetic polymer that is one of the most widely used polymers in controlled/directed drug delivery systems. This is due to its properties, such as biodegradability (hydrolysis to lactic acid and glycolic acid, which are metabolized by the body through the Krebs cycle), biocompatibility, and the ease of encapsulation of different types of drugs [21,49,50].

##### Lipid Nanoparticles

We can consider two categories of lipid nanoparticles: solid lipid nanoparticles (SLN) and nanostructured lipid carriers (NLC). SLN are matrix nanoparticles made of solid lipids dispersed in water or an aqueous surfactant solution. They have high physical stability, no organic solvents used in their manufacture, good biocompatibility and tolerability, and also allow a controlled drug release and prolong the nasal retention time due to their occlusive effect and adhesion to the mucosa. Despite these advantages, SLN have some drawbacks. These include a limited ability to solubilize hydrophilic molecules, low drug encapsulation efficiency (due to their crystalline structure), possible drug expulsion during storage due to the crystallization process, and an undesirable increase in particle size by agglomeration, which can lead to an immediate and unwanted drug release [33,36,42]. To improve some of these aspects, such as low stability, surfactants such as polyethylene glycol (PEG) can be used. PEG is a hydrophilic and biocompatible polymer that stabilizes nanoparticles and acts as a mucus penetration enhancer [21,51].

In turn, instead of having only solid lipids in their composition, NLC have a mixture of solid and liquid lipids that form an imperfect crystalline matrix into which drugs can be incorporated. This imperfect matrix increases the drug-loading capacity of the system and minimizes/avoids its immediate and undesired release during storage, thus overcoming these disadvantages of SLN. NLC are biodegradable and generally composed of physiological lipids, and therefore have low toxicity to the body and good tolerability. With this nanosystem, it is possible to achieve a higher drug encapsulation efficiency (compared to SLN), as hydrophobic molecules have a higher solubility in liquid lipids than in solid lipids. Nevertheless, the solubilization capacity of hydrophilic drugs is still low, which is a drawback of NLC [20,21,33].

##### Nanometric Emulsions

Regarding the nature of the external and internal phase(s), we can distinguish four types of nanometric emulsions: oil-in-water or water-in-oil (two phases) or oil-in-water-in-oil or water-in-oil-in-water (three phases). Oil-in-water nanometric emulsions, which are the most common, can solubilize and encapsulate hydrophobic drugs. A nanometric emulsion can improve drug stability and solubility and provide greater absorption due to the large surface area created by the small and numerous droplets [40,42,52].

Nanometric emulsions can also be classified according to other characteristics in microemulsions or nanoemulsions. Microemulsions (ME) are isotropic and thermodynamically stable colloidal dispersions. They are generally composed of oil, water, and surfactants, and have droplet sizes between 10 and 100 nm. The IN administration of an oil-in-water ME may allow direct transport to the brain due to its small droplet size and lipophilic nature. In turn, nanoemulsions are also colloidal systems of nanometric size, with droplet sizes between 20 and 200 nm, but they are thermodynamically unstable, which can lead to poor stability and drug release during storage. Similar to microemulsions, they also have an oil phase, an aqueous phase, and a surfactant. However, the latter is usually present in smaller quantities [21,38,41].

##### Nanogels

Nanogels are non-fluid colloidal or polymeric networks that increase their volume when in contact with a fluid, producing homogeneous solutions with low viscosity. These are defined as gel particles with a diameter of less than 100 nm. The use of nanogels is more effective than free drug administration (solution or other simple dispersion), since they have reported reduced toxicity, increased drug cellular uptake, high drug loading, and controlled drug release. This delivery system is effective for brain targeting as it leads to fast brain-drug absorption and has high biodegradability, biocompatibility, and hydrophilicity. It offers some advantages over other structures, such as the ability to encapsulate multiple molecules with different characteristics (both hydrophilic or hydrophobic) in the same formulation, leading to controlled drug release [53,54].

##### Liposomes

Liposomes are biocompatible and biodegradable vesicles composed of layers of phospholipids and cholesterol, enclosing one or more aqueous compartments. They may be unilamellar (smaller size) or multilamellar (larger size). Liposomes can transport hydrophilic and hydrophobic molecules due to their structural properties: hydrophilic drugs can be stored in the aqueous nucleus, whereas hydrophobic molecules can be dissolved in the lipid membrane. They have an overall good permeation capacity, including through the nasal mucosa, and can protect drugs from enzymatic degradation. Nevertheless, immunogenicity is a problem that needs to be solved, along with low encapsulation efficiency, to reduce the need for frequent administration. Compared to bigger liposomes, the smaller ones (generally neutral or positively charged) have a longer circulation time. Additionally, unmodified liposomes have a short circulation time and, therefore, are rapidly cleared, via systemic circulation, by the smooth endoplasmic reticulum cells [33,42,55].

Thus, to improve the circulation time of liposomes, modifications are made to their surface, such as coating them with PEG chains. In addition to PEG, it is common to use poloxamers, which are water-soluble, non-ionic polymers consisting of a triblock copolymer: a hydrophobic polypropylene glycol chain and two hydrophilic PEG chains. Suitable examples are poloxamer 407 and poloxamer 188, which have a high PEG content. Additionally, PEG reduces the viscosity of the nasal mucus and increases the penetration into the mucosa by interacting with lipid membranes and occlusion junctions, which is quite favorable for intranasal administration [21,42,54].

## 2. Successful Approaches to Increasing Brain Targeting and Bioavailability of Antidepressant and Anxiolytic Drugs

In general, the most common strategies for the increased brain targeting and bioavailability of antidepressant and anxiolytic drugs are: delivery via IN route; the use of several different types of nanosystems; and the complexation of drugs with compounds that allow them to be delivered more effectively to the desired site of action.

The most commonly studied classes of antidepressant drugs include selective serotonin reuptake inhibitors (SSRI), serotonin and noradrenaline reuptake inhibitors (SNRI), monoamine oxidase (MAO) inhibitors, and melatonin agonists. The effects of anxiolytic drugs, such as benzodiazepines, and natural products, antiepileptics, analgesics, and other drug classes on anxiety and depression have also been studied.

A summary of these molecules and their respective classifications is shown in Table 1, and a summary of the successful strategies that have been used to increase these drugs’ brain targeting and bioavailability is present in Table 2.

### 2.1. Antidepressant Drugs

#### 2.1.1. Agomelatine

Agomelatine is a melatonin MT1 and MT2 receptor agonist and a serotonin 5-HT2C receptor antagonist. It has low oral bioavailability, a high first-pass effect, and a short elimination half-life (2–3 h) [49]. The study conducted by Jani et al. [49] aimed to circumvent these obstacles by producing a polymeric NP using PLGA and poloxamer 407. The PLGA NP had a size of 116 nm, a PDI of 0.057, and a zeta potential (ZP) of −22.7 mV. In ex vivo drug permeation studies (goat nasal mucosa), the developed formulation had the highest permeability, compared to an agomelatine suspension, probably due to its reduced particle size and high homogeneity. In vivo pharmacodynamic studies were conducted using forced swim tests. In these tests, it was demonstrated that rats induced with depression were immobile during the five minutes that they were in the water. In contrast, rats in which PLGA NP with agomelatine was administered (IN route) showed a significant reduction in immobility time. Hence, the PLGA NP proved to have therapeutic efficacy, effectively delivering the drug to the brain.

#### 2.1.2. Selegiline

Selegiline is an MAO inhibitor and is a dose-dependent drug, which means that while a higher dose inhibits MAO-A and MAO-B, a lower dose only inhibits MAO-B. It has a high first-pass metabolism, low bioavailability, and a large number of adverse effects [56]. To overcome these problems, Singh et al. [56] developed thiolated chitosan NP (TCN) and unmodified chitosan NP (CNP) to enhance the intranasal delivery of selegiline. The particle size, PDI, and ZP of TCN were 215 nm, 0.057, and +17.06 mV, respectively. In terms of in vitro drug release, it was noted that at an early phase, within the first 2.5 h, the drug release from CNP was faster when compared to TCN. However, after 13 h, it was shown that the TCN had a cumulative release of 80%, whereas for the CNP this was only 68%, indicating an extended release by the TCN, but also a higher drug release at the end of the assay. In in vivo assays performed in rats, TCN significantly attenuated oxidative stress and restored mitochondrial complex activity. In addition, pharmacodynamic studies, which included immobility and locomotor activity tests, showed that, compared to CNP, which already resulted in more movement and less immobilized time, TCN had a more significant effect on improving the condition of the rats. Additionally, the sucrose preference test revealed that, at the beginning of the treatment, these rats had a decrease in water-with-sucrose intake, and an increase in intake at the end of the 14 days, during which the TCN were administered. Thus, the developed TCN seem to be promising candidates for IN administration in depression treatment.

### 2.2. Anxiolytic Drugs

#### 2.2.1. Buspirone

Buspirone is a drug that has proven to be effective as an anxiolytic. Nevertheless, it undergoes extensive first-pass metabolism by the cytochrome P3A4 in the liver and intestine, resulting in low oral bioavailability (approximately 4%) and a short half-life (2–3 h). It is currently only available on the market as an oral tablet, and multiple daily doses are required because only a small amount of the drug reaches the intended therapeutic site of action [57,58,59].

Patil et al. [57] investigated whether the intranasal administration of buspirone could transport the drug directly to the brain, thereby increasing the cerebral bioavailability of the drug. Therefore, an intranasal buspirone solution was formulated with chitosan and β-cyclodextrins and compared with the intravenous administration of an aqueous solution of the drug. This study confirmed the increase in bioavailability and targeted delivery to the brain with the developed formulation IN delivery (DTP 76%), demonstrating the efficacy of the nasal route, as well as the incorporation of cyclodextrins and chitosan into the formulation, which increased not only drug solubility, but also its permeation and transport to the brain.

In turn, Bari et al. [58], developed TCN for the intranasal delivery of buspirone to the brain. Comparative studies were performed with CNP, and IN and IV drug solutions. In vitro drug release studies showed that while the percentage of buspirone release from the drug solution was almost complete after 2 h, the drug release in that same time from the CNP and TCN was only 51.8 and 46.6%, respectively. Hence, the developed nanosystems showed a controlled drug release. Furthermore, the ex vivo drug permeation study (porcine nasal mucosa) showed that TCN had a higher percentage of drug permeation across the nasal mucosa than all the other evaluated formulations. These results were probably due to the increased mucoadhesion capacity of the TCN, derived from the formation of a strong ionic bond between the amine groups of thiolated chitosan and the negatively charged sites on mucosal epithelial cell membranes, resulting in the transient opening of the tight junctions. In vivo behavioral studies were also carried out on rats to assess the formulation’s anxiolytic activity, namely the maze test, in which part of the maze is exposed to light (open arm), and the other is not (closed arm). Characteristically, anxious animals tend to remain in the closed arm (as do anxious people, when they show a desire to be isolated and in dark places). This study concluded that rats treated with TCN spent more time in the open arm of the maze when compared to other groups, showing promising therapeutic efficacy. Furthermore, the brain maximum drug concentration (Cmax) of buspirone after the IN administration of TCN (797.46 ng/mL) was higher than that obtained with the IN and IV solution (417.77 ng/mL and 384.15 ng/mL, respectively). Additionally, the drug-targeting efficiency (DTE%) and direct transport percentage (DTP%) for the IN TCN were 79% and 96%, respectively, and hence brain drug transport was considered to be quite high, which further proved the developed formulation’s potential.

In another study, by Bshara et al. [59], the aim was to develop a buspirone mucoadhesive ME for IN administration, to improve its bioavailability and deliver high drug concentrations to the brain. This ME was prepared using chitosan, hydroxypropyl cyclodextrin, isopropyl myristate, Tween^®^ 80 (polysorbate derived from sorbitan esters), propylene glycol, and water. The results showed that mucoadhesive ME are suitable for IN administration, since the drug’s brain bioavailability in in vivo studies was significantly increased compared to an IN solution, reaching peak plasma concentrations within 15 min. The DTE% and DTP% values obtained with the developed IN ME were 86.6% and 88%, respectively. From these results, it may be concluded that a buspirone mucoadhesive ME can contribute to a reduction in the dose and frequency of administration and possibly to an increase in the therapeutic effect in anxiety treatment.

#### 2.2.2. Clobazam

Clobazam is a benzodiazepine derivative with an active metabolite, norclobazam. Both are partial GABA receptor agonists, meaning they bind allosterically to the GABA-A receptor. Clobazam is a lipophilic molecule and is highly bound to plasma proteins, which increases its systemic distribution in fat tissue, causing adverse effects such as gastrointestinal disturbances, muscle spasms, and an irregular heartbeat. The prolonged oral use of this drug leads to an accumulation of its active metabolite in the body (10 times higher) which can cause toxicity. Additionally, it can lead to dependence when used for prolonged periods of time [60]. Florence et al. [60] aimed to develop and characterize a mucoadhesive clobazam ME to assess the drug’s transport to the brain, thereby decreasing the drug’s systemic distribution and consequently its potential to cause adverse effects. The developed mucoadhesive ME consisted of Carbopol^®^ 940P, Capmul^®^ MCM (glyceryl mono and dicaprate), Acconan^®^ CC6, Tween^®^ 20 (polysorbate derived from sorbitan esters), and distilled water. Carbopol was used due to its capacity to increase paracellular transport by opening tight junctions in apical cells, resulting in higher drug absorption. The non-mucoadhesive microemulsion had a viscosity value of 7.73 cP, and the mucoadhesive microemulsion had a viscosity value of 25.8 cP, and hence the addition of Carbopol to the formulation led to a viscosity increase, which was expected, since, aside from being a mucoadhesive polymer, Carbopol also has viscosifying properties. The obtained droplet size was 20 nm, the PDI was 0.181, and the ZP was −15 mV. In the pharmacokinetic study in mice, the obtained brain/blood ratio was higher with IN administration of the mucoadhesive ME, with greater and longer drug retention at the site of action when compared to a clobazam non-mucoadhesive ME (same composition, but no Carbopol). In addition, the systemic distribution of the drug was generally lower with IN administration, and there was a higher accumulation in the brain, as intended. Hence, this study showed that the clobazam ME with the addition of a mucoadhesive agent delivered the drug rapidly and effectively to the mice’s brains. This could provide an alternative to intravenous administration in the treatment of anxiety.

### 2.3. Anxiolytic and Antidepressant Drugs

#### 2.3.1. Venlafaxine

Venlafaxine (VLF) belongs to the SNRI therapeutic class and has low oral bioavailability and a short half-life (4–5 h). It is also a P-glycoprotein substrate and is therefore pumped out of the brain, reducing its bioavailability in this organ. Its effectiveness also depends on its continuous presence at the site of action for a prolonged period of time [19,61,63].

To overcome these limitations, Haque et al. [61] investigated the usefulness of IN administration of VLF-loaded QT NP to improve their delivery to the brain, compared to IV infusion. The VLF nanoparticles were formulated using the ionic gelation technique, containing QT, tripolyphosphate (TPP), and acetic acid. The obtained particle size was 167 nm, with a PDI of 0.367 and a ZP of +23.83 mV. The VLF in vitro drug release from the developed NP showed two phases, with an initial burst release, corresponding to 44.3%, in the first 2 h, followed by a controlled release of the drug, over 24 h. An ex vivo permeation study, conducted on porcine nasal mucosa, showed that QT VLF NP were able to permeate the membrane three times more than a VLF solution. In vivo pharmacokinetic studies showed the effective brain drug targeting of the NP through the IN route (Figure 5A), when compared to the IV route (Figure 5B), reaching high DTE% (508.59%) and DTP% (80.34%) values. This proves that QT-NPs were highly efficient in delivering VLF to the brain when administered intranasally, which could be greatly due to their mucoadhesive properties. Additionally, in in vivo pharmacodynamic studies, QT NP significantly increased the total swimming and climbing time and decreased the immobility time of the mice (compared to the control groups), which further proved the nanosystem’s therapeutic potential.

The same authors presented another study [62], where they also developed VLF-loaded NP for IN administration, but this time instead of QT they used AG. These NP had a particle size of 174 nm, a PDI of 0.391, a pH between 5.7 and 6.12, and a ZP of +37 mV. It was observed in pharmacokinetic in vivo studies that the drug concentration in the brain increased when the formulation was administered via IN when compared to the IV route. The brain/blood ratios (Figure 5C) obtained with the administration of the IN AG NP were generally higher than those obtained from the IV and IN drug solution, being equal to 0.1091, 0.0293, and 0.0700, respectively, at the final time point (480 min). The results also showed that the plasma concentration of VLF (Figure 5D) decreased when the drug was administered within the NP through IN delivery, showing the promise of higher safety, and that brain drug levels (Figure 5E) were also substantially higher with the IN NP when compared to the other groups, showing higher efficacy. The DTE% was almost double for the VLF AG NP when compared to the IN VLF solution. This proves that the mucoadhesive nanoparticulate delivery system enables an increased amount of drug to reach the brain. In vivo pharmacodynamic studies further demonstrated the efficacy of VLF AG NP delivered via the IN route to treat depression, since in forced swim tests there was a significantly reduced immobility time after IN NP administration, proving it to be more effective than all other groups.

Another study, by Cayero-Otero et al. [63], developed NP composed of PLGA and polyvinyl alcohol, with a particle size of 206 nm, a PDI of 0.041, and a ZP of −26.5 mV. These NP showed a higher capacity to reach the brain after IN administration than functionalized NP, with a specific transferrin receptor agonist peptide on their surface. This was explained by the fact that functionalized NP are transported to the brain by receptor-mediated endocytosis, which takes longer than simple NP, which are transported by facilitated transport.

In a different study, Zhao et al. [64] developed two solutions containing two different conjugates: venlafaxine-glucose (VLF-G) and thiamine disulfide system with venlafaxine-glucose conjugated (VLF-TDS-G). Type 1 glucose transporters (GLUT1) are responsible for transporting glucose to the brain and are present in the BBB. Therefore, this study aimed to investigate the binding of VLF to a glucose molecule in order to originate the drug’s facilitated transport to the brain, through the BBB, after IV administration. A thiamine disulfide system was introduced into the VLF-G complex to prevent premature VLF release. In in vivo pharmacokinetics studies, the VLF-TDS-G conjugate significantly increased the brain VLF concentration, compared to the VLF-G conjugate, demonstrating good brain targeting. Thus, the VLF-TDS-G conjugate is promising for the brain delivery of intravenously administered drugs.

#### 2.3.2. Duloxetine

Duloxetine is a drug that belongs to the SNRI class. It undergoes a high hepatic first-pass metabolism and has low oral bioavailability (only 50%) [65]. To overcome these problems, Alam et al. [65] developed duloxetine SLN for IN administration. The SLN were composed of glyceryl monostearate, Capryol^®^ PGMC, bile salts (sodium taurocholate), Pluronic^®^ F-68, and mannitol. The authors did not characterize the formulation, which at least raises questions about the homogeneity and particle size of the developed nanosystem, characteristics that can influence the absorption and bioavailability of the drug. Nevertheless, the SLN’s performance was evaluated in in vivo studies. The IN administration of the developed SLN resulted in a six times higher concentration of duloxetine in the brain when compared to the IV administration of a drug solution. Hence, drug delivery to the brain was much lower with IV administration, with only small amounts of the drug being detected, due to loss in the systemic circulation and the hepatic first-pass effect. On the contrary, IN administration proved to be quite efficient in making the drug reach the brain, with the developed duloxetine SLN leading to high DTP (86.80%) and DTE (758%) values. The developed IN SLN also had a better brain targeting efficiency than an IN drug solution (DTP 65%, DTE 287%), which further proved the potential of the developed nanosystem for duloxetine IN brain delivery.

#### 2.3.3. Paroxetine

Paroxetine belongs to the SSRI class, has low oral bioavailability (less than 50%), and undergoes an extensive first-pass effect. It also has a high potential for drug–drug interactions, especially when used in polytherapy [66]. To overcome some of these problems, Silva et al. [66] investigated the efficacy of the direct delivery of paroxetine to the brain in combination with another drug, borneol. Borneol is a natural compound that has already been shown to be effective in opening channels in the BBB. Therefore, researchers believe that borneol acts as a modulator of the ABC transporters by competitively inhibiting the P-gp and tight junction proteins, thereby reducing the efflux of drugs that are their substrates [46,66,73]. It is a reversible process characterized by transient and rapid BBB penetration. This co-administration of borneol with paroxetine works in a dose- and time-dependent manner. Thus, paroxetine was incorporated together with borneol into NLC containing Lauroglycol™ 90, Precirol^®^ ATO 5, Tween^®^ 80, and water. The developed nanosystem had a particle size of 160 nm, a PDI of 0.273, and a ZP of +11 mV. In vitro permeation studies in RPMI 2650 cells (human nasal cells) showed a 2.57-fold increase in drug permeation when the developed NLC was compared with a paroxetine suspension. The IN administration of borneol and paroxetine-loaded NLC to mice allowed the researchers to obtain a 63% higher brain exposure (as measured by the area under the concentration vs. time curve, AUC) than an IV injection. Additionally, the administration of a paroxetine IN solution only increased brain drug exposure by 49%, which further proves the superiority of the developed NLC. It was also observed that drug encapsulation reduced pulmonary exposure to the drug, which may contribute to a reduction in adverse effects. Thus, the developed NLC seem to be a good strategy to increase the IN delivery of lipophilic drugs, such as paroxetine, to the brain.

### 2.4. Other Drug Classes

#### 2.4.1. Baicalein

Baicalein (BA) is the most therapeutically active natural flavonoid compound found in the dried roots of *Scutellaria baicalensis georgi*, with proven beneficial CNS and immunological effects [70]. Given these properties, Chen et al. [70] prepared SLN for baicalein encapsulation, composed of glyceryl monostearate, poloxamer 188, and 1,2-dipalmitoyl-sn-glycerol-3-phosphocholine. The developed nanosystems were additionally functionalized with the prolyl-glycyl-proline (PGP) peptide (PGP-BA-SLN) or without (BA-SLN) and had a ZP between −13 and −14 mV, and a pH of 5.5. The changes in lactate dehydrogenase (LDH) concentrations were studied in rats after the intraperitoneal administration of the developed formulations, as depression often involves hormonal changes leading to the negative regulation of LDH, resulting in the accumulation of lactate in the brain (from lactic acid fermentation). Both PGP-BA-SLN and BA-SLN enhanced the effect of BA in reducing LDH release compared to a drug solution. There was no statistically significant difference between the treatments. Pharmacodynamic tests were also carried out, and the results showed that both types of SLN reduced immobility time, increased climbing time, and increased swimming time in rats, with PGP-BA-SLN having a more significant effect than BA-SLN. Additionally, the brain drug distribution was determined, and the results showed that BA was found in higher concentrations in the basolateral amygdala after the administration of the developed SLN, a brain region associated with emotional and psychiatric disorders. All these results lead to the conclusion that modifying SLN with PGP may be beneficial in brain drug targeting, and hence producing effective antidepressant effects.

#### 2.4.2. Icariin

Icariin is the main active constituent of the dried aerial parts of the *Epimedium brevicornum Maxim* plant. It has antitumor, cardiovascular, osteogenic, neuroprotective, and antidepressant effects. Nevertheless, it is poorly absorbed after oral administration [71]. To tackle these problems, Xu et al. [71] developed a thermosensitive nano-hydrogel (icariin-NGSTH) for the IN delivery of icariin, in order to enhance the antidepressant activity and bioavailability of this drug. A mixture of poloxamer 188 and poloxamer 407 was used to form an in situ thermosensitive hydrogel (Figure 6A), with the purpose of extending the drug’s release time and therefore increasing its bioavailability, by giving it more time to be absorbed in the nasal cavity. The nano-hydrogel was also composed of Tween^®^ 80 and Span^®^ 80. The formulation had a particle size of 71 nm, a PDI of 0.50, and a ZP of −19 mV. Regarding pharmacokinetics, the in vivo distribution of the developed icariin nano-hydrogel in the brain of mice and rats was monitored after IN administration. The rats and mice treated with the developed IN nanosystem were compared to animals chronically treated with fluoxetine, an antidepressant of known effectiveness. After the IN administration of the icariin-NGSTH, fluorescence could be observed in the brain after 5 min, which became stronger after 30 min (Figure 6B). These results indicate that the drug reached the brain quickly, therefore producing a potentially fast therapeutic effect. This was further confirmed by the pharmacodynamic tests, since the IN administration of the developed formulation reduced the immobility time of the mice as well as other depressive behaviors. Moreover, the dose of icariin-NGSTH administered was only one-fifth of the dose of fluoxetine, and one-tenth of the dose of icariin administered in the form of a solution, demonstrating that the developed nanosystem indeed had promising antidepressant effects. Additionally, IL-6, a pro-inflammatory cytokine, is increased in depression because of neuroinflammatory processes. Hence, it was also observed that the icariin-NGSTH, at a lower dose, reduces the expression of IL-6 (Figure 6C), and also regulates the level of testosterone (Figure 6D).

#### 2.4.3. Tramadol

Tramadol is a central analgesic with a low affinity for opioid receptors, but its active metabolite has a higher affinity for these receptors. It is also characterized by the inhibition of noradrenaline and serotonin receptors [53]. The study performed by Kaur et al. [53] aimed to evaluate the efficacy of tramadol NP administered through the IN route. The developed NP were composed of Pluronic^®^ F-127 and TPP. The obtained particle size was 152 nm, with a PDI of 0.143 and a ZP of +31 mV. Pharmacodynamic tests were performed in rats, namely the forced swimming, locomotor activity, immobility, body weight variation, and glucose preference tests. The tests were conducted after the chronic administration of the developed formulations, and showed a significant decrease in immobility time and an increase in locomotor activity and body weight in the rats treated with the NP, compared to the control groups. Biochemical parameters were also measured in the animals’ brains, and it was concluded that the developed NP could reduce nitrite and malondialdehyde levels more significantly than a drug solution. Additionally, GSH and catalase levels increased when the NP were administered.

#### 2.4.4. Edaravone

Edaravone is a neuroprotective drug that scavenges free radicals and protects the neuronal membranes from oxidative damage. For this reason, it is used to treat amyotrophic lateral sclerosis. However, as oxidative processes also play a huge role in depression and anxiety, it has recently gained interest as a treatment for these pathologies [68]. Qin et al. [68] developed edaravone liposomes modified with the RGD peptide (arginine-glycine-aspartate), to be administered by intraperitoneal administration (Figure 7A). This peptide has the advantage of binding to leukocytes and thus penetrating the BBB in cases of neuroinflammation, which is present in depression. The cyclic RGD peptide (cRGD) has been shown to have an even higher affinity for the receptors expressed on the surface of leukocytes. In this study, a bacterial endotoxin (lipopolysaccharide, LPS) was administered to rats to impair their social behavior, since LPS leads to the production of inflammatory cytokines, such as IL-6, which causes neuropsychiatric disorders and depression. Three types of liposomes were prepared: functionalized edaravone liposomes composed of edaravone, cRGD, soy phosphatidylcholine, 1,2-dipalmitoyl-sn-glycero-3-phosphocholine, and cholesterol; non-functionalized edaravone liposomes consisting only of edaravone and cholesterol; and functionalized liposomes with no edaravone, composed of cRGD peptide, soy phosphatidylcholine, 1,2-dipalmitoyl-sn-glycero-3-phosphocholine, and cholesterol. The results showed that the functionalized edaravone liposomes were more effective than the other formulations in increasing the mobility of the rats in the forced swim test (Figure 7B). Notably, this formulation also significantly attenuated the levels and reduced the secretion of IL-1β (Figure 7C) and IL-6 (Figure 7D), suggesting that edaravone may suppress the production of pro-inflammatory cytokines by scavenging free radicals.

#### 2.4.5. Carbamazepine

Carbamazepine (CBZ) is an antiepileptic and anxiolytic drug which exerts its anxiolytic activity by modulating the adenosine-mediated neurotransmitters to modify postsynaptic ionic currents. Its low aqueous solubility and extensive hepatic metabolism result in slow absorption after oral administration. Furthermore, the absorption is variable and depends on the drug’s dissolution rate in the gastrointestinal fluids [67]. To improve CBZ’s brain delivery, Khan et al. [67] investigated the potential of CBZ NLC by intraperitoneal administration. The optimized formulation had a particle size of 97.7 nm, a PDI of 0.27, and a ZP of −22 mV. The in vitro drug release assay showed that, when compared to a CBZ dispersion, the developed CBZ NLC exhibited a biphasic release profile, with a faster drug release in the first 4 h (Figure 8A). The achieved CBZ concentration after formulation administration was also assessed in mice, and the CBZ NLC significantly increased the AUC of CBZ (520.4 µg.h/mL) in the brain compared to the CBZ dispersion (244.9 µg.h/mL). Additionally, the administration of the CBZ NLC resulted in higher plasma (Figure 8B) and brain (Figure 8C) concentrations at each time point. Furthermore, at the same time points, the brain concentrations of CBZ-TLNs were higher than their plasma concentrations, which represents the potential for a favorable efficacy vs. safety profile. The anxiolytic effect of the CBZ NLC, assessed in in vivo pharmacodynamic studies, was demonstrated and proved to be higher than that produced with the CBZ dispersion (Figure 8D–F). Additionally, the CBZ NLC showed better results than diazepam treatment, which is a known effective anxiolytic drug.

#### 2.4.6. Riluzole

Riluzole is a drug that works by reducing the release of glutamate in the synaptic cleft, making it difficult for glutamate receptors to be activated, thus protecting the dopaminergic neurons. It also helps to reduce oxidative stress and improve memory. All these factors are related to its potential anxiolytic activity. However, this drug undergoes extensive first-pass metabolism by CYP1A2, which hinders its clinical efficacy [69]. To circumvent these problems, Nabi et al. [69] developed chitosan NP, since it is a polymer that has been described as having the ability to enhance NP mucoadhesive strength and increase drug absorption, thereby improving drug delivery to the brain. Furthermore, as already mentioned, chitosan has the ability to facilitate the transient opening of tight junctions, as well as prevent the degradation of the encapsulated drug. The developed chitosan NP were also functionalized with transferrin, which allows them to cross the BBB by transcytosis. This happens because transferrin receptors are widely distributed on the brain capillary endothelial cells, which are involved in the transcytosis process. Therefore, transferrin can freely cross the unimpaired regions of the BBB. The developed NP were characterized and had a particle size of 207 nm and PDI of 0.406. In in vivo pharmacodynamic tests, haloperidol was administered to rats included in the study before the administration of any formulation, to induce anxiety symptoms, as haloperidol leads to neuronal degradation, which produces oxidative stress. The results showed that, compared to the control groups (treated with only haloperidol or with haloperidol plus a riluzole IN solution), the rats treated with IN administration of the developed NP showed a greater improvement in terms of neurological effects. Additionally, biochemically the group treated with haloperidol only showed decreased GSH levels and increased malondialdehyde, whereas the treatment group showed the reverse. In addition, the TBARS levels were found to be lower in the rats treated with the developed NP, with more significant effects than those provoked by the administration of a drug solution. Furthermore, in in vivo pharmacokinetic studies the developed NP led to a higher brain drug uptake by the IN route. The use of transferrin was shown to be an additional advantage, increasing the efficiency of drug delivery to the brain. Hence, the developed NP may have a high therapeutic potential for drug delivery via the IN route.

#### 2.4.7. Berberine

Berberine is an alkaloid found in various medicinal plants, such as *Coptis Chinensis*, and is mainly isolated from its bark and roots. It has known medicinal properties, including antioxidant, anti-inflammatory, and hepatoprotective effects. Several studies show that berberine can inhibit MAO-A, the enzyme responsible for the degradation of noradrenaline and serotonin. Despite all these properties, it has low bioavailability [74,75,76,77]. Hence, Wang et al. [72] investigated the incorporation of berberine into a thermosensitive gel, to be administered through the IN route. This gel exhibited properties such as high fluid absorption and low surface tension, resulting in good biocompatibility and high drug incorporation capacity. These properties are due to the polymers included in its composition, namely poloxamers 407 and 188, non-ionic copolymers consisting of a hydrophobic polyoxypropylene chain and two hydrophilic polyoxyethylene chains. Before the preparation of the hydrogel system, the authors improved the solubility of berberine by using cyclodextrins. Hence, first a drug–cyclodextrin inclusion complex was prepared, and then this complex was incorporated into the hydrogel. The intermolecular interactions between the cyclodextrins and the poloxamers P407 and P188 resulted in the formation of a supramolecular matrix. This formulation was then evaluated in in vivo pharmacokinetic studies in rats: one group received the inclusion complex incorporated into a thermosensitive hydrogel through the IN route at a lower dose (0.15 mg/kg); and the other group received the inclusion complex in an aqueous dispersion (no incorporation into a gel) through the intragastric route at a higher dose (5 mg/kg). The inclusion complex incorporated into a thermosensitive hydrogel proved to be a more effective and faster treatment than the inclusion complex in an aqueous dispersion, even though it was administered at a dose three times lower (higher C_max_ and AUC values). Furthermore, in a study where reserpine was first administered to rats (reserpine is known to exhaust the monoamine neurotransmitters at the synapses, resulting in depression-like behavior), the IN administration of the hydrogel caused a significant increase in the levels of serotonin, noradrenaline, and dopamine in the rat hippocampus and striatum, showing superior efficacy compared to all the other groups.

### 2.5. General Analysis

#### 2.5.1. Formulation Characteristics

Considering that most surveyed articles opted for drug incorporation into a nanosystem, it is necessary to make a general analysis of the reported formulation characteristics. These characteristics mostly included the particle size, PDI, ZP, pH, and viscosity.

Most of the articles mentioned the particle size, PDI, and ZP. These parameters are very important as they influence the absorption and consequently the bioavailability of the encapsulated drugs. The particle size must be in the nanometer range for more effective drug delivery. In addition, formulations with a smaller particle size are more likely to be able to penetrate the mucous membranes [21]. The PDI is directly related to the particle size, being defined as the standard deviation of the particle diameter distribution divided by the mean particle diameter [78]. It is usually used to estimate formulation homogeneity, which is very important in drug pharmacokinetics. A higher value means that the formulation is heterogeneous, which can lead to pharmacokinetic irregularities and variability in the therapeutic outcomes. It is usually recommended that the PDI must be less than 0.5, which was the case in all the articles that analyzed this parameter [20].

The nanosystem’s surface charge can also affect the BBB and nasal mucosa penetration [46]. It is usually determined by ZP measurement. The nasal mucosa has a negative charge, so positively charged moieties are more likely to interact with the nasal mucosa through electrostatic forces, therefore increasing the residence time and the formulation’s adhesion to the nasal epithelium. For this reason, many nanocarriers are positively charged. This will lead to a potential increase in the bioavailability of the delivered molecules [21]. Conventionally, nanosystems with a ZP between −10 and +10 mV are considered neutral, while nanoparticles with a ZP higher than +30 mV or less than −30 mV are considered strongly cationic or strongly anionic, respectively. In this review, only two articles presented formulations with a ZP above +30 mV or below −30 mV, which means that although most of the formulations were considered neutral, they still managed to permeate the membranes with some efficiency [79].

Most authors opted for IN administration of the developed formulations. Thus, these preparations had to meet certain conditions adapted to this specific administration route, namely a pH between 5.0 and 6.0 (nasal mucosa’s pH) to avoid irritation or harm. The articles that reported pH values mentioned results between 4.62 and 7.00, which are appropriate for compatibility with the nasal mucosa’s physiology. Viscosity is also a relevant factor in IN administration, since the higher a formulation’s viscosity, the longer the formulation will remain in contact with the nasal mucosa, and hence the drug will have more time to undergo absorption [28]. Nevertheless, pH and viscosity were the two characteristics least mentioned in most articles, which may raise the question of the suitability of the preparations for IN administration.

#### 2.5.2. In Vivo Pharmacokinetics and Pharmacodynamics

In vivo pharmacodynamic and/or pharmacokinetic studies were conducted to evaluate the performance of the developed formulations after administration. The used animal models were either rats or mice.

Pharmacokinetic studies were performed to quantify the drug that was delivered to the brain, and also the part that was not. Several different analytical methods were used to quantify the blood/plasma samples (representative of systemic distribution) and brain samples (representative of the desired site of action), namely: high-performance liquid chromatography, liquid chromatography coupled with mass spectrometry, gas chromatography coupled with mass spectrometry, scintigraphy, and fluorescence. C_max_, which represents the maximum drug concentration that was reached during the study in a specific biological tissue, and AUC, which represents the change in drug concentration over time, are the main parameters that can be assessed in these types of study. From the brain and blood/plasma AUC values specific ratios can be calculated in order to assess the brain-targeting efficiency of the developed formulations: DTE% and DTP%. DTE% is a measure of brain drug transport via IN delivery versus IV delivery. DTE% values above 100% indicate that the drug is more efficiently transported to the brain by the IN route when compared to the IV route. It can be calculated as follows:$$\mathrm{DTE}\%=\frac{\left(\frac{\mathrm{AUC}\;\mathrm{brain}}{\mathrm{AUC}\;\mathrm{plasma}}\right)\mathrm{IN}}{\left(\frac{\mathrm{AUC}\;\mathrm{brain}}{\mathrm{AUC}\;\mathrm{plasma}}\right)\mathrm{IV}}\;\times \;100$$

DTP% is a similar ratio, in the way that it also compares IN and IV administrations, but it represents the proportion of the drug that is transported directly to the brain. DTP% values above 0% indicate that the drug is transported by neuronal pathways (direct pathways). It can be calculated as follows:$$\mathrm{DTP}\%=\frac{B_{\mathrm{IN}}{-\;B}_{X}}{B_{\mathrm{IN}}}{\;\times \;100\;\mathrm{where}\;B}_{X}=\frac{B_{\mathrm{IV}}}{P_{\mathrm{IV}}}{\;\times \;P}_{\mathrm{IN}}$$

Higher DTE% and DTP% values indicate that the drug’s IN administration has better brain targeting efficiency than IV administration [20].

As for the performed pharmacodynamic tests, these included the forced swimming, locomotor activity, and sucrose preference tests. These are the main tests that are performed for the assessment of depression and/or anxiety. The forced swim test is one of the most commonly used tests to verify the immobility time of the animal. In general, the depressed animal has a longer immobility time than the healthy animal. After administering the various formulations to groups of animals in which depression was induced, a significant reduction in immobility time was obtained. This allows us to conclude that the administration of the developed formulations was successful in making the drugs reach the brain, and hence having an antidepressant effect. Another important test, reported in some of the articles included in this review, is the locomotor activity test. Depressed animals are more likely to show increased immobility compared to healthy ones, or those receiving antidepressant/anxiolytic treatment [53]. The third most reported test is the sucrose preference test, which evaluates sucrose consumption during treatment with antidepressants or anxiolytics compared to control groups. The results show that sucrose consumption does not change in treated animals but increases in untreated animals [80]. In general, all the analyzed studies showed a positive evolution in the animals’ behavior, demonstrating that the developed treatment is more effective than other administration routes and/or formulations.

#### 2.5.3. Overview and Future Prospects

After analyzing all the brain targeting strategies of different drugs, we can conclude that it is imperative to have as much information as possible to achieve a fast and effective brain drug delivery. In this way, we must know the nanosystem’s characteristics, such as particle size, PDI, and ZP, as well as the characteristics of the final preparation, such as pH and viscosity. Among the analyzed articles, the most commonly used nanosystems were polymeric and lipid nanoparticles, which proved to be more effective in delivering the drug to the brain through the nasal cavity. Additionally, an IN formulation for the treatment of depression has already been developed and approved by the United States Food and Drug Administration in 2019. It is a nasal spray containing esketamine, with the brand name Spravato^®^. It is used for the treatment of major depression along with suicidal ideation when patients demonstrate resistance to oral antidepressants [81,82,83]. This can be an open door for the further development and approval of IN antidepressant and anxiolytic drugs. While there are currently no preparations containing nanosystems for the treatment of these diseases in the pharmaceutical market, multiple studies have now proven their efficacy. Hence, further studies still need to be performed in order to assess the true potential of nanosystems containing antidepressant and anxiolytic drugs for IN administration, especially regarding their efficacy and safety in clinical trials. Additionally, the scale-up difficulties that may arise from preparations containing nanosystems should be assessed and improved, so that one day the reported promising experimental results can originate a marketed formulation, and these preparations can be an option for improved depression and anxiety treatment.

## 3. Conclusions

Drug molecule functionalization has proven to be a promising alternative for antidepressant and/or anxiolytic drug modification when systemic administration is required, since with the right ligands it could lead to increased drug transport through the BBB, leading to higher brain drug concentrations. Nevertheless, the IN route has proven to be an excellent alternative to systemic routes, such as oral and intravenous administrations, as it can allow the drugs to be transported directly to the brain through neuronal transport without having to pass through the BBB. This administration route has proven to lead to higher efficacy (increased brain targeting and bioavailability) and safety (a reduction of systemic drug distribution). Furthermore, formulating drugs into nanosystems has proven to increase the therapeutic efficacy in animal models of these diseases, being especially relevant through IN administration. Hence, IN administration of antidepressant and anxiolytic drugs has been demonstrated to be a suitable alternative to the treatments currently available on the pharmaceutical market for the treatment of depressive and anxiety disorders. Hence, it is essential than in the future further studies are conducted, so that these formulations could one day reach the market and ensure an improvement in the quality of life of patients suffering from these pathologies.

## Figures and Tables

**Figure 1 pharmaceutics-15-00998-f001:**
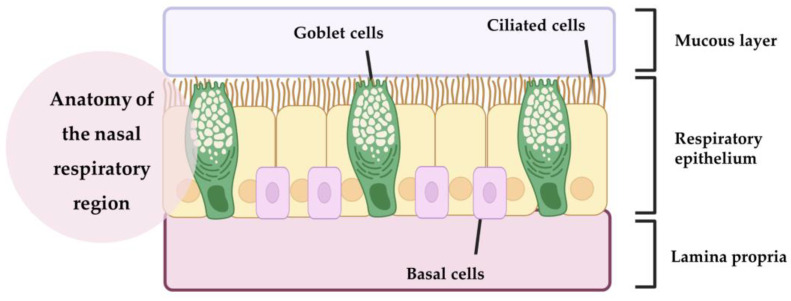
Anatomy of the nasal respiratory region (produced with BioRender).

**Figure 2 pharmaceutics-15-00998-f002:**
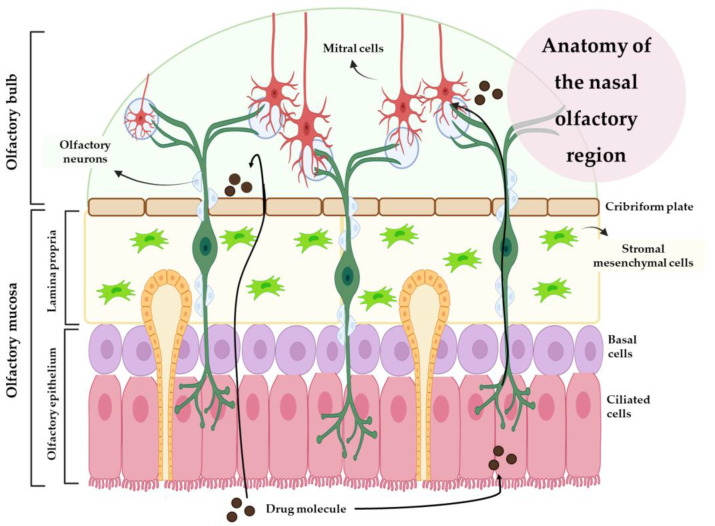
Anatomy of the nasal olfactory region (produced with BioRender).

**Figure 3 pharmaceutics-15-00998-f003:**
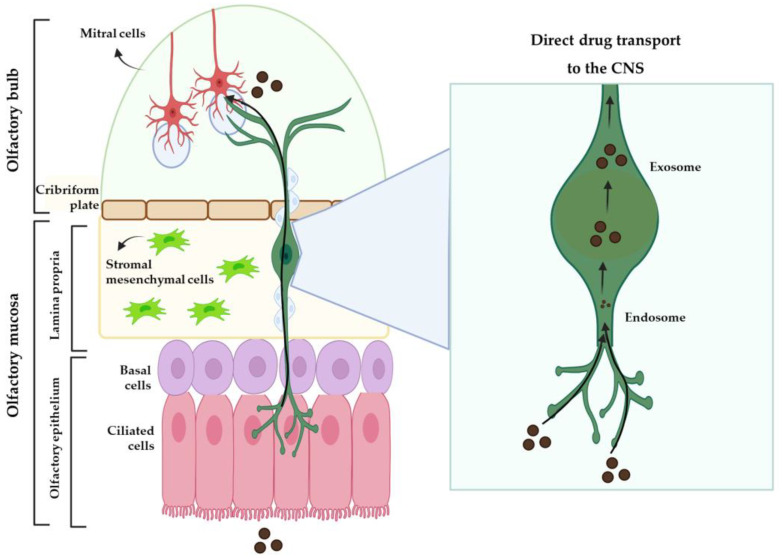
Extracellular mechanism of drug transport to the brain from the nasal cavity (produced with BioRender).

**Figure 4 pharmaceutics-15-00998-f004:**
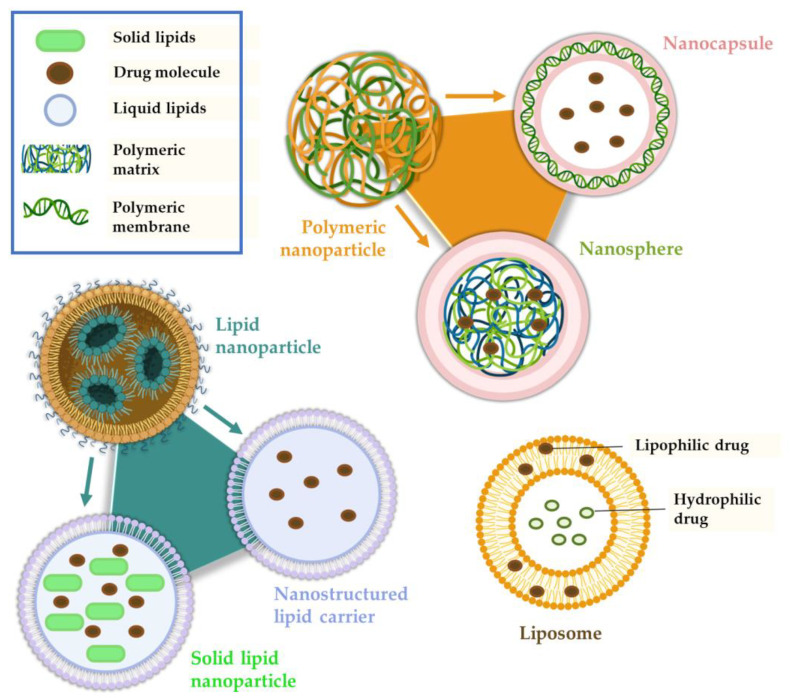
Main therapeutic nanosystem types (produced with BioRender).

**Figure 5 pharmaceutics-15-00998-f005:**
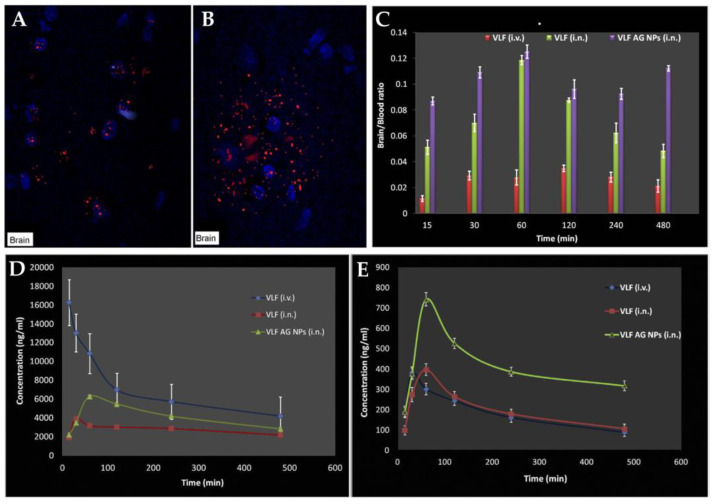
Confocal laser scanning microscopy rat brain images 120 min after IV administration (**A**) and IN administration (**B**) of the developed VLF QT NP, where blue fluorescence represents the brain cells and red fluorescence represents the drug (adapted from Haque et al. [61], reproduced with permission from Elsevier (license number 5495991231429)). Brain/blood ratios (**C**), plasma concentrations (**D**), and brain concentrations (**E**) of VLF after IV administration of a VLF solution (VLF (i.v.)), IN administration of a VLF solution (VLF (i.n.)), or IN administration of VLF AG NP (VLF AG NPs (i.n.)); adapted from Haque et al. [62], reproduced with permission from Elsevier (license number 5497041230698).

**Figure 6 pharmaceutics-15-00998-f006:**
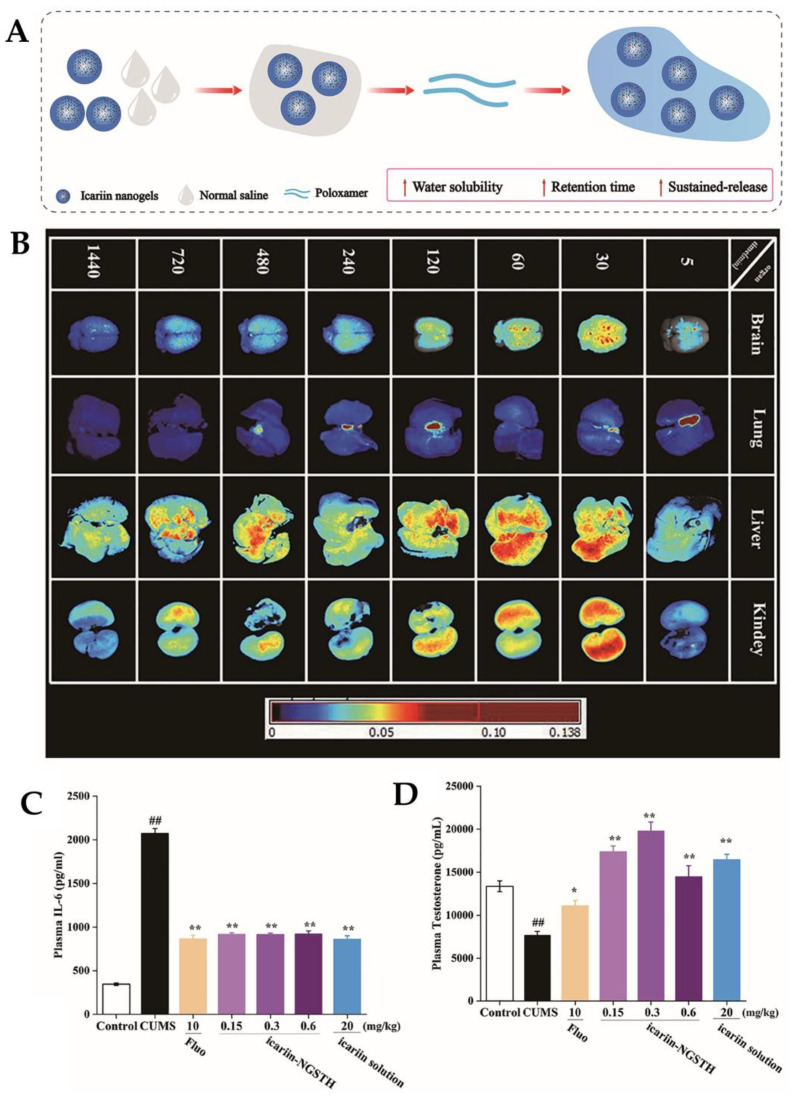
(**A**) Schematic representation of the developed icariin-NGSTH. (**B**) In vivo distribution of rhodamine B-labeled icariin-NGSTH. (**C**) IL-6 concentration in rat plasma after administration. (**D**) Testosterone concentration in rat plasma after administration. * *p* < 0.05, ## or ** *p* < 0.01; NGSTH—thermosensitive nano-hydrogel. Adapted from Xu et al. [71], reproduced with permission from Elsevier (license number 5496000528811).

**Figure 7 pharmaceutics-15-00998-f007:**
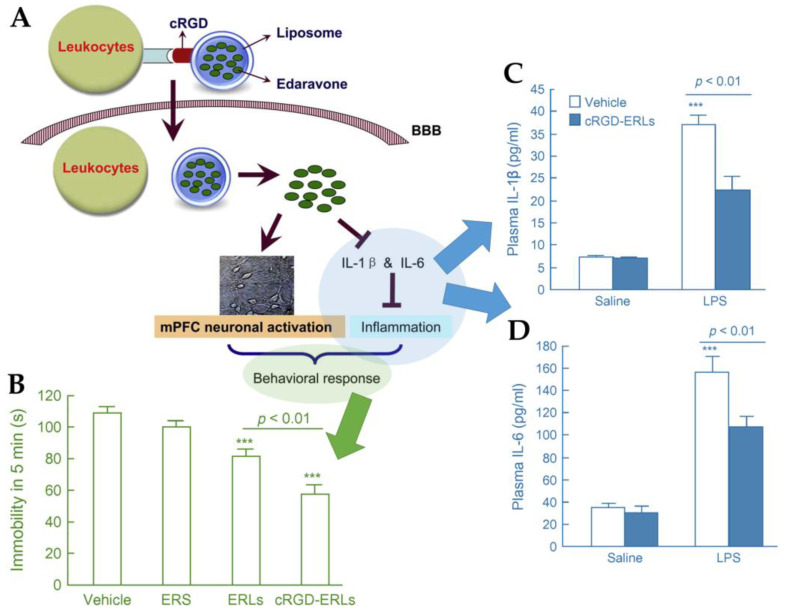
(**A**) Schematic representation of the developed functionalized edaravone liposomes. (**B**) Immobility time in the forced swim test after administration. Plasma IL-1β (**C**) and IL-6 (**D**) after administration. BBB—blood–brain barrier; cRGD—cyclic RGD (arginine-glycine-aspartate) peptide; cRGD-ERLs—functionalized edaravone liposomes; ER—edaravone; ERLs—non-functionalized edaravone liposomes; ERS—edaravone solution; IL—interleukin; LPS—lipopolysaccharide. Adapted from Qin et al. [68], reproduced with permission from Elsevier (license number 5496000926348).

**Figure 8 pharmaceutics-15-00998-f008:**
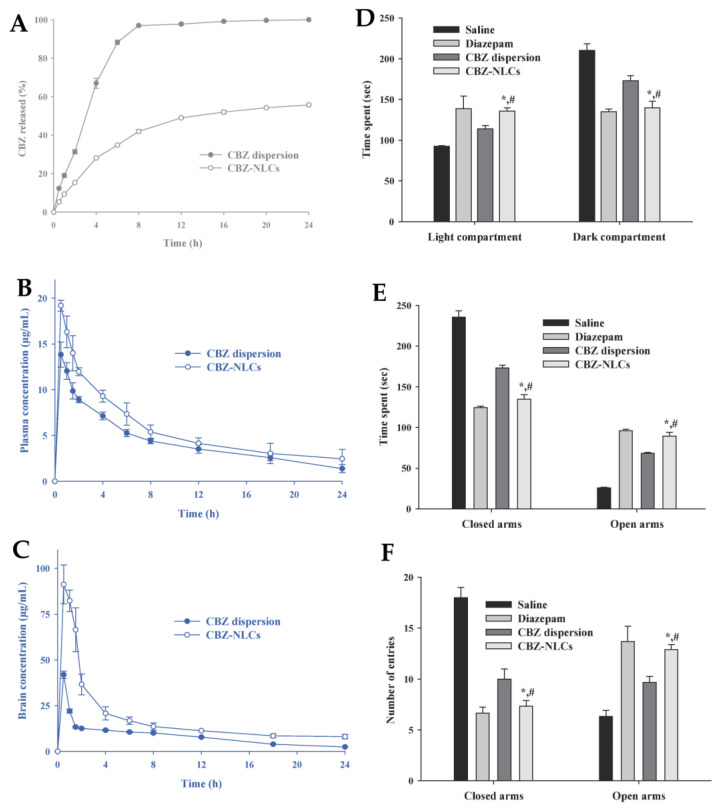
(**A**) In vitro drug release profile of a carbamazepine dispersion (“CBZ dispersion”) and the developed NLC (“CBZ-NLCs”). CBZ plasma (**B**) and brain (**C**) concentration versus time curves after the intraperitoneal administration of a carbamazepine dispersion (“CBZ dispersion”) and the developed NLC (“CBZ-NLCs”). (**D**) to (**E**) The results of the administration of several formulations in a light-dark box mice model (**D**) and elevated plus maze mice model (**E**,**F**). * and # *p* < 0.01. Adapted from Khan et al. [67], reproduced with permission from Elsevier (license number 5496001134122).

**Table 1 pharmaceutics-15-00998-t001:** Summary of the drug molecules included in this review, including their approved drug classification and known action mechanism(s).

Drug Name	Drug Classification	Action Mechanism(s)	References
Agomelatine	Antidepressant	Melatonin MT1 and MT2 receptor agonist and a serotonin 5-HT2C receptor antagonist	[49]
Selegiline		MAO inhibitor	[56]
Buspirone	Anxiolytic	Serotonin 5-HT1A receptor agonist	[57,58,59]
Clobazam		Partial GABA receptor agonist	[60]
Venlafaxine	Antidepressant and anxiolytic	Serotonin and noradrenaline reuptake inhibitor	[61,62,63,64]
Duloxetine		Serotonin and noradrenaline reuptake inhibitor	[65]
Paroxetine		Selective serotonin reuptake inhibitor	[66]
Carbamazepine	Antiepileptic	Modulation of adenosine-mediated neurotransmitters	[67]
Tramadol	Analgesic	Opioid agonist and serotonin and noradrenaline reuptake inhibitor	[53]
Edaravone	Amyotrophic lateral sclerosis treatment	Free radical scavenger	[68]
Riluzole		Glutamate antagonist	[69]
Baicalein	Natural product	NA	[70]
Icariin		NA	[71]
Berberine		MAO inhibitor	[72]

NA—not available.

**Table 2 pharmaceutics-15-00998-t002:** Summary of the successful strategies that have been used to increase brain drug targeting and bioavailability in the treatment of depressive and anxiety disorders.

Drug Name	General Strategy	Nanosystem Type (When Applicable)	References
Agomelatine	Nanosystems and intranasal administration	Polymeric nanoparticles	[49]
Selegiline			[56]
Buspirone			[58]
		Microemulsion	[59]
	Intranasal administration	-	[57]
Clobazam	Nanosystems and intranasal administration	Microemulsion	[60]
Venlafaxine	Nanosystems and intranasal administration	Polymeric nanoparticles	[61]
			[62]
			[63]
	Drug molecule functionalization	-	[64]
Duloxetine	Nanosystems and intranasal administration	Solid lipid nanoparticles	[65]
Paroxetine		Nanostructured lipid carriers	[66]
Carbamazepine	Nanosystems		[67]
Tramadol	Nanosystems and intranasal administration	Polymeric nanoparticles	[53]
Edaravone	Nanosystems	Liposomes	[68]
Riluzole	Nanosystems and intranasal administration	Polymeric nanoparticles	[69]
Baicalein	Nanosystems	Solid lipid nanoparticles	[70]
Icariin	Intranasal administration	-	[71]
Berberine		-	[72]

## Data Availability

Not applicable.

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
