# Peer review of "Nanosystems, Drug Molecule Functionalization and Intranasal Delivery: An Update on the Most Promising Strategies for Increasing the Therapeutic Efficacy of Antidepressant and Anxiolytic Drugs"

_pharmaceutics, 2023, doi:10.3390/pharmaceutics15030998_

Round 1

Reviewer 1 Report

This manuscript presents the treatment of depression and anxiety disorders currently dominated by oral medications. Then the author introduced three main strategies have been used to improve brain drug targeting: intranasal route of administration, nanosystems for drug encapsulation and drug molecule functionalization. The description of the article is very detailed but needs to be more intuitive. It would be better to use tables or graphs to summarize important information.

1.     The authors summarize anxiolytic and antidepressant medications in a large amount of text, suggesting that the authors add a table describing current anxiolytic and antidepressant medications.

2.     Are there currently any nanosystems, functionalization of drug molecules, and intranasal drug delivery that have been applied in clinical treatment for depression and anxiety? If so, please give an example. If not, please introduce the limitations of these methods and the aspects that need further research in the future.

Author Response

R: We thank the reviewer for their insightful comments. All suggestions and corrections have been taken into account (changes marked in blue in the revised version of the manuscript), and a point-by-point answer in given below.

  1. The authors summarize anxiolytic and antidepressant medications in a large amount of text, suggesting that the authors add a table describing current anxiolytic and antidepressant medications.

R: We thank the reviewer for their comment and agree that a summary of the included drugs was necessary, and hence have added a Table with this content, including the drug molecule name, approved classification, and main action mechanism(s) (page 9, Table 1, marked in blue).

  1. Are there currently any nanosystems, functionalization of drug molecules, and intranasal drug delivery that have been applied in clinical treatment for depression and anxiety? If so, please give an example. If not, please introduce the limitations of these methods and the aspects that need further research in the future.

R: We thank the reviewer for their comment. There is currently one intranasal formulation in the market, as stated from lines 854 to 860: “Additionally, an IN formulation for the treatment of depression has already been developed and approved by the United States Food and Drug Administration in 2019. It is a nasal spray containing esketamine, with the brand name is Spravato®. It is used for the treatment of major depression along with suicidal ideation, when patients demonstrate resistance to oral antidepressants”. As for drug molecule functionalization and/or nanosystems, to the best of our knowledge there are currently no preparations of the kind in the pharmaceutical market. The already existing comment on future perspectives for this type of formulations has now been further developed and completed, from lines 860 to 868 (marked in blue).

Reviewer 2 Report

1.    The article has useful information and well written but needs some tables for important findings of the literatures and some figures as well as further updates for the references.

2.    In page no.2 Authors need to cite references in these sentences.In 2020, this disease affected about 16% of the world's population.”

3.    In page no.2 Authors need to cite references in these sentences.The BBB also has a biochemical layer with high levels of efflux transport proteins, such as P-glycoprotein (Pgp) and multidrug-resistant protein-1, as well as the expression of many metabolic enzymes, which limit brain drug uptake.”

4.    In page no.4 Authors should rewrite the following sentences.

In addition, since IV drug delivery to the brain is largely influenced by the drug's plasma half-life, the extent of metabolism, the degree of non-specific binding to plasma proteins, and the permeability of the compound across the BBB and into peripheral tissues, intranasal (IN) administration has been presented as a promising alternative, that has gained increasing interest.”

5.    There is no table in this review article. Authors should provide tabular data for reader's attention. E.g., Authors should provide a table for different types of Nanosystems which have been used for brain targeting for Antidepressant and Anxiolytic drugs. Authors should also provide a table for Successful approaches to increasing brain targeting and bioavailability of antidepressant and anxiolytic drugs. 

Author Response

R: We thank the reviewer for their comments. All suggestions and corrections have been taken into account (changes marked in blue in the revised version of the manuscript), and a point-by-point answer in given below.

  1. The article has useful information and well written but needs some tables for important findings of the literatures and some figures as well as further updates for the references.

R: We thank the reviewer for their comment, changes have been made regarding these issues according to all reviewers comments. Namely, one table summarizing the drug molecules included in this review, including their approved drug classification and known action mechanism(s) has been added to page 9 (Table 1); and another table summarizing the successful strategies that have been used to increase drug brain targeting and bioavailability in the treatment of depressive and anxiety disorders has also been added, on page 10 (Table 2).

  1. In page no.2 Authors need to cite references in these sentences. “In 2020, this disease affected about 16% of the world's population.”

R: We thank the reviewer for their comment, the reference has now been added (marked in blue).

  1. In page no.2 Authors need to cite references in these sentences. “The BBB also has a biochemical layer with high levels of efflux transport proteins, such as P-glycoprotein (Pgp) and multidrug-resistant protein-1, as well as the expression of many metabolic enzymes, which limit brain drug uptake.”

R: We thank the reviewer for their comment, the adequate references have been added (marked in blue).

  1. In page no.4 Authors should rewrite the following sentences.

“In addition, since IV drug delivery to the brain is largely influenced by the drug's plasma half-life, the extent of metabolism, the degree of non-specific binding to plasma proteins, and the permeability of the compound across the BBB and into peripheral tissues, intranasal (IN) administration has been presented as a promising alternative, that has gained increasing interest.”

R: We thank the reviewer for their comment, it was indeed a long and maybe confusing sentence, and has now been rewritten and divided into two sentences, for better reader comprehension (marked in blue).

  1. There is no table in this review article. Authors should provide tabular data for reader's attention. E.g., Authors should provide a table for different types of Nanosystems which have been used for brain targeting for Antidepressant and Anxiolytic drugs. Authors should also provide a table for Successful approaches to increasing brain targeting and bioavailability of antidepressant and anxiolytic drugs.

R: We thank the reviewer for their comment, a Table has now been added summarizing the strategies that have been used for brain targeting of antidepressant and anxiolytic drugs, including the specification of each nanosystem category. This has been done in Table 2, added to page 10 (marked in blue).

Round 2

Reviewer 2 Report

The authors have answered all the queries very well.